# Coordination Polymers Constructed from Semi-Rigid *N,N′*-Bis(3-pyridyl)terephthalamide and Dicarboxylic Acids: Effect of Ligand Isomerism, Flexibility, and Identity

**Chia-Jou Chen** [1], **Chia-Ling Chen** [1], **Yu-Hsiang Liu** [1], **Wei-Te Lee** [1], **Ji-Hong Hu** [1], **Pradhumna Mahat Chhetri** [2,*] and **Jhy-Der Chen** [1,*]

1   Department of Chemistry, Chung-Yuan Christian University, Chung-Li 320, Taiwan; jelly4112@gmail.com (C.-J.C.); jialing41666@gmail.com (C.-L.C.); g10963021@cycu.edu.tw (Y.-H.L.); lku23y230230@gmail.com (W.-T.L.); joejoe860831@gmail.com (J.-H.H.)
2   Department of Chemistry, Amrit Campus, Tribhuvan University, Kathmandu 44600, Nepal
*   Correspondence: mahatp@gmail.com (P.M.C.); jdchen@cycu.edu.tw (J.-D.C.); Tel.: +886-3-265-3351(J.-D.C.)

**Abstract:** Reactions of the semi-rigid *N,N′*-bis(3-pyridyl)terephthalamide (**L**) with divalent metal salts in the presence of dicarboxylic acids afforded $[Cd(\mathbf{L})_{0.5}(1,2\text{-}BDC)(H_2O)]_n$ (1,2-$H_2$BDC = benzene-1,2-dicarboxylic acid), **1**, $\{[Cd(\mathbf{L})_{1.5}(1,3\text{-}BDC)(H_2O)]\cdot 5H_2O\}_n$ (1,3-$H_2$BDC = benzene-1,3-dicarboxylic acid), **2a**, $\{[Cd(1,3\text{-}BDC)(H_2O)_3]\cdot 2H_2O\}_n$, **2b**, $\{[Cd(\mathbf{L})_{0.5}(1,4\text{-}BDC)(H_2O)_2]\cdot H_2O\}_n$ (1,4-$H_2$BDC = benzene-1,4-dicarboxylic acid), **3**, and $[Cu(\mathbf{L})_{0.5}(5\text{-tert-IPA})]_n$ (5-tert-IPA = 5-tert-butylbenzene-1,3-dicarboxylic acid), **4**, which have been structurally characterized by single crystal X-ray diffraction. Complexes **1** and **3** are two-dimensional (2D) layers with the **bey** and the **hcb** topologies, and **2a** and **2b** are one-dimensional (1D) ladder and zigzag chain, respectively, while **4** shows a 3-fold interpenetrated three-dimensional (3D) net with the **cds** topology. The structures of these coordination polymers containing the semi-rigid **L** ligands are subject to the donor atom positions and the identity of the dicarboxylate ligands, which are in marked contrast to those obtained from the flexible bis-pyridyl-bis-amide ligands that form self-catenated nets. The luminescence of **1** and **3** and thermal properties of complexes **1**, **3**, and **4** are also discussed.

**Keywords:** coordination polymers; semi-rigid ligand; ligand isomerism; ligand flexibility

## 1. Introduction

Coordination polymers (CPs) constructed from metal salts and spacer ligands are coordination compounds with repeating coordination entities extending in 1, 2, or 3 dimensions [1]. As a result of their interesting structures and properties, CPs have been extensively studied by scientists during recent years [2,3]. Apart from metal ions, a wise choice of the spacer ligands involving the neutral and the auxiliary polycarboxylate anions are important in determining the structural types of CPs in a mixed ligand system, which are also subject to counter ions, solvents, temperature, metal to ligand ratio, etc.

The chemistry of CPs containing flexible bis-pyridyl-bis-amide (bpba) ligands has been investigated extensively during recent year [4]. The ligand-isomerism of the flexible bpba as well as the auxiliary dicarboxylate ligands have been found important in determining the structural diversity [5–8]. Cd(II) perchlorate CPs constructed from the flexible and isomeric *N,N′*-di(2-pyridyl)adipoamide, *N,N′*-di(3-pyridyl)adipoamide and *N,N′*-di(4-pyridyl)adipoamide exhibit a 1D zigzag chain, a 2D pleated sheet, and a 3D entangled framework, respectively [5], while the Cd(II) CPs comprising *N,N′*-di(3-pyridyl)adipoamide ligands are directed by the isomeric 1,2-, 1,3-, and 1,4-benzenedicarboxylate ligands, resulting in a 1D ladder chain, a 2D layer with loops and a 3D self-catenated net, respectively [6]. Reactions of the flexible *N,N′*-di(3-pyridyl)subero-amide with Cu(II) salts in the presence of the isomeric 1,2-, 1,3-, and 1,4-phenylenediacetic

acids afforded CPs showing a 3D net with the 3,5T1 topology, a 5-fold interpenetrated 3D net of **cds** topology, and a 1D self-catenated network [7], while reactions of $N,N'$-di(3-pyridyl)suberoamide and Cd(II) salts with the isomeric phenylenediacetic acids afforded a loop-like 1D chain, a self-catenated net with the $(6^5 \cdot 8)$ topology and a 2D layer with the **sql** topology, respectively [8]. Probably, the self-catenated CPs can be achieved by the manipulation of the ligand-isomerism of the auxiliary polycarboxylate ligands.

We are investigating the CPs comprising the semi-rigid bpba and the dicarboxylate ligands, in an attempt to elucidate the effect of the rigidity of the neutral spacer on the structures of CPs supported by the dicarboxylate ligands [9]. In this report, $N,N'$-bis(3-pyridyl)terephthalamide (**L**), Figure 1, and divalent metal salts were used to react with the isomeric benzene-dicarboxylic acids as well as 5-tert-butylbenzene-1,3-dicarboxylic acid to investigate the effect of the donor atom position and the identity of the polycarboxylate ligands on the structural diversity of CPs with semi-rigid spacer. The syntheses and structures of $[Cd(L)_{0.5}(1,2\text{-BDC})(H_2O)]_n$ (1,2-$H_2$BDC = benzene-1,2-dicarboxylic acid), **1**, $\{[Cd(L)_{1.5}(1,3\text{-BDC})(H_2O)] \cdot 5H_2O\}_n$ (1,3-$H_2$BDC = benzene-1,3-dicarboxylic acid), **2a**, $\{[Cd(1,3\text{-BDC})(H_2O)_3] \cdot 2H_2O\}_n$, **2b**, $\{[Cd(L)_{0.5}(1,4\text{-BDC})(H_2O)_2] \cdot H_2O\}_n$ (1,4-$H_2$BDC = benzene-1,4-dicarboxylic acid), **3**, and $[Cu(L)_{0.5}(5\text{-tert-IPA})]_n$ (5-tert-IPA = 5-tert-butylbenzene-1,3-dicarboxylic acid), **4**, form the subject of this report. Thermal and luminescent properties of the available complexes are also discussed.

**Figure 1.** Structures of **L**, 5-tert-butylbenzene-1,3-dicarboxylic acid (5-tert-$H_2$IPA), benzene-1,2-dicarboxylic acid (1,2-$H_2$BDC), benzene-1,3-dicarboxylic acid (1,3-$H_2$BDC), and benzene-1,4-dicarboxylic acid (1,4-$H_2$BDC).

## 2. Materials and Methods

### 2.1. General Procedures

Elemental analyses were performed on a PE 2400 series II CHNS/O (PerkinElmer Instruments, Shelton, CT, USA). Solid state IR spectra were measured on a JASCO FT/IR-460 plus spectrometer (JASCO, Easton, MD, USA). Powder X-ray diffraction (PXRD) patterns were carried out on a Bruker D2 PHASER diffractometer (Bruker Corporation, Karlsruhe, Germany). Solid state emission spectroscopy was done using a Hitachi F-4500 spectrometer (Hitachi, Tokyo, Japan). TGA curves were obtained form an SII Nano Technology Inc. TGA/DTA 6200 analyzer (Seiko Instruments Inc., Torrance, CA, USA) in 30–800 °C under a nitrogen atmosphere.

## 2.2. Materials

The reagent $Cu(OAc)_2 \cdot H_2O$ was purchased from SHOWA Co. (Saitama, Japan), 5-tert-butylbenzene-1,3-dicarboxylic acid (5-tert-$H_2$IPA) from Aldrich Co. (Wyoming, IL, USA), benzene-1,2-dicarboxylic acid (1,2-$H_2$BDC), benzene-1,3-dicarboxylic acid (1,3-$H_2$BDC) and benzene-1,4-dicarboxylic acid (1,4-$H_2$BDC) from ACROS Co. (Pittsburgh, PA, USA), and $Cd(OAc)_2 \cdot 2H_2O$ from Fisher Scientific Co.(Pittsburgh, PA, USA). $N,N'$-bis(3-pyridyl)terephthalamide (**L**) was prepared according to published procedures [10].

## 2.3. Preparations

### 2.3.1. $[Cd(L)_{0.5}(1,2\text{-}BDC)(H_2O)]_n$, **1**

A mixture of $Cd(OAc)_2 \cdot H_2O$ (0.027 g, 0.10 mmol), **L** (0.032 g, 0.10 mmol), benzene-1,2-dicarboxylic acid (1,2-$H_2$BDC) (0.017 g, 0.10 mmol), and 10 mL of 0.01 M NaOH solution in water was sealed in a 23 mL Teflon-lined stainless steel autoclave which was heated under autogenous pressure to 120 °C for two days and then the reaction system was cooled to room temperature at a rate of 2 °C per hour. Colorless crystals suitable for single-crystal X-ray diffraction were obtained which were washed with water and dried under vacuum. Yield: 0.022 g (49%). Anal. Calcd for $C_{17}H_{13}CdN_2O_6$ (MW = 453.69): C, 45.00; H, 2.89; N, 6.17%. Found: C, 44.99; H, 3.05; N, 6.16%. FT-IR ($cm^{-1}$): 3500(w), 3068(m), 1676(s), 1612(s), 1550(s), 1488(s), 1430(s), 1378(s), 1336(m), 1301(s), 1198(m), 1108(m), 800(m), 760(m), 731(m), 696(m).

### 2.3.2. $\{[Cd(L)_{1.5}(1,3\text{-}BDC)(H_2O)] \cdot 5H_2O\}_n$, **2a**, and $\{[Cd(1,3\text{-}BDC)(H_2O)_3] \cdot 2H_2O\}_n$, **2b**

Prepared by following the similar procedures for **1** except that 1,3-$H_2$BDC was reacted at 90 °C. Different colorless crystals with poor crystallinity were obtained, which were difficult to be separated manually. Careful test of the crystals showed that two complexes $\{[Cd(L)_{1.5}(1,3\text{-}BDC)(H_2O)] \cdot 5H_2O\}_n$, **2a**, and $\{[Cd(1,3\text{-}BDC)(H_2O)_3] \cdot 2H_2O\}_n$, **2b**, can be found, which were structurally characterized by using single crystal X-ray crystallography. Although the X-ray crystallography data of **2a** are humble, its structure can be clearly identified.

### 2.3.3. $\{[Cd(L)_{0.5}(1,4\text{-}BDC)(H_2O)_2] \cdot H_2O\}_n$, **3**

Prepared by following the similar procedures for **1** except that benzene-1,4-dicarboxylic acid (1,4-$H_2$BDC) (0.017 g, 0.10 mmol) was used. Yield: 0.026 g (38%). Anal. Calcd for $C_{17}H_{15}CdN_2O_7 \cdot H_2O$ (MW = 489.74): C, 41.69; H, 3.50; N, 5.72%. Anal. Calcd for $C_{17}H_{15}CdN_2O_7$ (without cocrystallized water molecule) (MW = 453.69): C, 45.00; H, 2.89; N, 6.17%. Found: C, 44.25; H, 3.35; N, 6.85%. FT-IR ($cm^{-1}$): 3677(w), 3303(m), 1673(m), 1608(m), 1552(s), 1488(s), 1386(s), 1332(s), 1295(s), 1203(m), 1120(m), 1012(w), 885(w), 838(s), 806(m), 746(m), 698(m), 646(m), 528(w).

### 2.3.4. $[Cu(L)_{0.5}(5\text{-tert-IPA})]_n$, **4**

Prepared as described for **1** except $Cu(OAc)_2 \cdot H_2O$ (0.020 g, 0.10 mmol), **L** (0.032 g, 0.10 mmol), 5-tert-$H_2$IPA (0.021 g, 0.10 mmol) and 10 mL of 0.01 M NaOH solution in water were used. Yield: 0.014 g (32%). Anal. Calcd for $C_{21}H_{19}CuN_2O_5$ (MW = 442.92): C, 56.71; H, 4.31; N, 6.30%. Found: C, 56.08; H, 3.99; N, 6.46%. FT-IR ($cm^{-1}$): 3271(m), 3203(m), 3118(m), 3066(m), 2970(m), 1680(s), 1625(s), 1583(s), 1543(s), 1479(m), 1444(s), 1421(s), 1378(s), 1331(s), 1305(m), 1279(s), 1238(m), 1189(m), 1112(m), 1053(w), 1016(w), 865(w), 806(m), 773(m), 748(m), 721(s), 692(s), 634(w).

The experimental powder X-ray patterns of **1**, **3,** and **4** match quite well with their simulated ones, indicating the bulk purities, Figures S1–S3.

## 2.4. X-ray Crystallography

A Bruker AXS SMART APEX II CCD diffractometer (graphite-monochromated MoK$\alpha$ = 0.71073 Å) was used to collect the diffraction data of complexes **1**–**4** [11]. Data reduction was performed by standard methods with use of well-established computational proce-

dures, involving Lorentz and polarization corrections and empirical absorption correction based on "multi-scan". Heavier atom positions were located by the direct method and the other atoms were found in alternating difference Fourier maps and least-square refinements. The hydrogen atoms except those of the water molecules were added by using the HADD command in SHELXTL 6.1012 [12]. Table 1 lists the crystal data for **1**, **2b**, **3**, and **4**, while those of **2a** are provided in the Supplementary Materials.

**Table 1.** Crystal data for complexes **1**–**4**.

| | **1** | **2b** | **3** | **4** |
|---|---|---|---|---|
| Formula | $C_{17}H_{13}CdN_2O_6$ | $C_8H_{14}CdO_9$ | $C_{17}H_{17}CdN_2O_8$ | $C_{21}H_{19}CuN_2O_5$ |
| Formula weight | 453.69 | 366.59 | 489.72 | 442.92 |
| Crystal system | Monoclinic | Triclinic | Triclinic | Monoclinic |
| Space group | $P2_1/c$ | $P\bar{1}$ | $P\bar{1}$ | $C2/c$ |
| a, Å | 14.6784(10) | 10.2076(3) | 6.34570(10) | 21.600(3) |
| b, Å | 5.8992(3) | 11.1103(3) | 9.4314(2) | 10.6511(15) |
| c, Å | 19.8917(11) | 12.6791(3) | 16.7255(3) | 17.041(2) |
| $\alpha$, ° | 90 | 102.1594(10) | 100.8415(12) | 90 |
| $\beta$, ° | 107.498(4) | 102.2797(10) | 95.6385(12) | 100.715 |
| $\gamma$, ° | 90 | 103.5718(10) | 104.0750(11) | 90 |
| V, Å$^3$ | 1642.74(17) | 1313.41(6) | 942.66(3) | 3852.2(9) |
| Z | 4 | 4 | 2 | 8 |
| $D_{calc}$, Mg/m$^3$ | 1.834 | 1.854 | 1.725 | 1.527 |
| F(000) | 900 | 728 | 490 | 1824 |
| $\mu$(Mo K$_\alpha$), mm$^{-1}$ | 1.368 | 1.697 | 1.206 | 1.170 |
| Range(2$\theta$) for data collection, deg | 2.91 to 52.00 | 3.42 to 56.63 | 4.70 to 56.63 | 3.84 to 56.52 |
| Independent reflections | 3173 [R(int) = 0.0794] | 6502 [R(int) = 0.0306] | 4703 [R(int) = 0.0344] | 4756 [R(int) = 0.0847] |
| Data/restraints/parameters | 3173/0/241 | 6502/0/348 | 4703/0/269 | 4756/492/285 |
| quality-of-fit indicator [c] | 1.004 | 1.003 | 1.034 | 1.059 |
| Final R indices [I > 2$\sigma$(I)] [a,b] | R1 = 0.0436, wR2 = 0.0561 | R1 = 0.0305, wR2 = 0.0965 | R1 = 0.0303, wR2 = 0.0596 | R1 = 0.0589, wR2 = 0.1310 |
| R indices (all data) | R1 = 0.0886, wR2=0.0638 | R1 = 0.0331, wR2 = 0.0994 | R1 = 0.0418, wR2=0.0633 | R1 = 0.0950, wR2=0.1486 |

[a] $R_1 = \Sigma||F_o| - |F_c||/\Sigma|F_o|$. [b] $wR_2 = [\Sigma_W(F_o^2 - F_c^2)^2/\Sigma w(F_o^2)^2]^{1/2}$. $w = 1/[\sigma^2(F_o^2) + (ap)^2 + (bp)]$, $p = [max(F_o^2 \text{ or } 0) + 2(F_c^2)]/3$. a = 0.0195, b = 0.0000 for **1**; a = 0.0624, b = 1.6811 for **2b**; a = 0.0276, b = 0.1920 for **3**; a = 0.0634, b = 0.0000 for **4**. [c] quality-of-fit = $[\Sigma_W(|F_o^2| - |F_c^2|)^2/(N_{observed} - N_{parameters})]^{1/2}$.

## 3. Results

### 3.1. Structure of *1*

A single-crystal X-ray diffraction analysis shows that **1** crystallizes in the monoclinic space group $P2_1/c$. The asymmetric unit consists of one Cd(II) cation, one half **L** ligand, one 1,2-BDC$^{2-}$ ligand, and one coordinated water molecule. Figure 2a depicts a drawing showing the coordination environment of Cd(II), which is seven-coordinated by six oxygen atoms from three independent 1,2-BDC$^{2-}$ ligands, one water molecule [Cd-O = 2.291(3) − 2.524(3) Å] and one pyridyl nitrogen atom from the **L** ligands [Cd-N = 2.273(4) Å], resulting in a distorted pentagonal bipyramidal geometry. The metal atoms are linked by the 1,2-BDC$^{2-}$ ligands to form 1D linear chains with various dinuclear units and the distances between the two Cd(II) metal centers are 5.0180(2) and 3.9650(2) Å for the large and small rings, respectively, Figure 2b. The 1D chains are further linked together by the **L** ligands to afford a 2D layer. If the Cd(II) cations are defined as 4-connected nodes and the 1,2-BDC$^{2-}$ cations as 3-connected nodes, while the **L** ligands as linkers, the structure of complex **1** can be regarded as a 3,4-connected 2D net with the $(4^2.6^3.8)(4^2.6)$-**bey**; 3,4L83 topology, Figure 2c, determined by using ToposPro program [13].

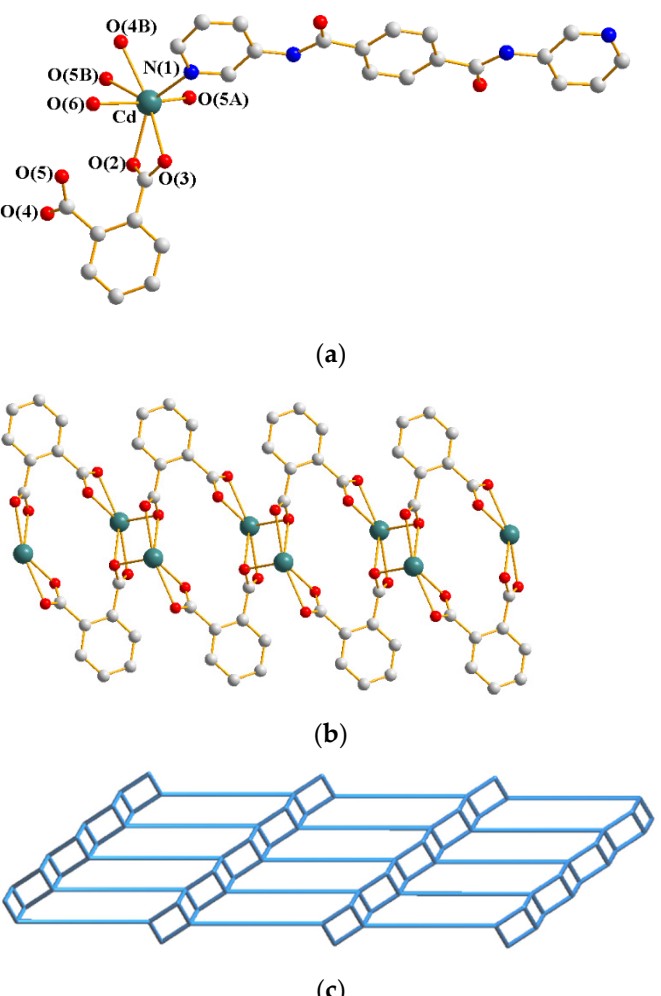

**(a)**

**(b)**

**(c)**

**Figure 2.** Figure 2. (**a**) Coordination environment of Cd(II) ion in **1**. Symmetry transformations used to generate equivalent atoms: (A) x, y + 1, z (B) −x + 1, −y + 1,− z + 1 (C) x, y − 1, z (D) −x + 2, −y + 4, −z + 1. (**b**) A drawing showing the 1D chain linked by the 1,2-BDC$^{2-}$ ligands. (**c**) A drawing showing the 2D layer with the **bey** topology.

### 3.2. Structures of **2a** and **2b**

Crystals of **2a** conform to the triclinic space group $P\bar{1}$. The asymmetric unit consists of one Cd(II) cation, one and a half **L** ligands, one 1,3-BDC$^{2-}$ ligand, one coordinated water molecule, and five cocrystallized water molecules. Figure 3a depicts a drawing showing the coordination environment about Cd(II), which is six-coordinated by three oxygen atoms from two 1,3-BDC$^{2-}$ ligands [Cd-O = 2.017(6) − 2.220(6) Å], one coordinated water molecule [Cd-O = 2.074(6) Å], and two pyridyl nitrogen atoms from the two **L** ligands [Cd-N = 2.193(6) and 2.225(7) Å], resulting in a distorted octahedral geometry. The metal atoms are linked by the 1,3-BDC$^{2-}$ ligands to form 1D ladder chains with bidentate bridging **L** as rungs and monodentate **L** dangling on the two supporting sides, Figure 3b. Interestingly, the 1D ladder chains are further reinforced by the **L** ligands through the π–π interactions (3.74, 3.86, and 3.73 Å) to afford a 2D layer, Figure 3c.

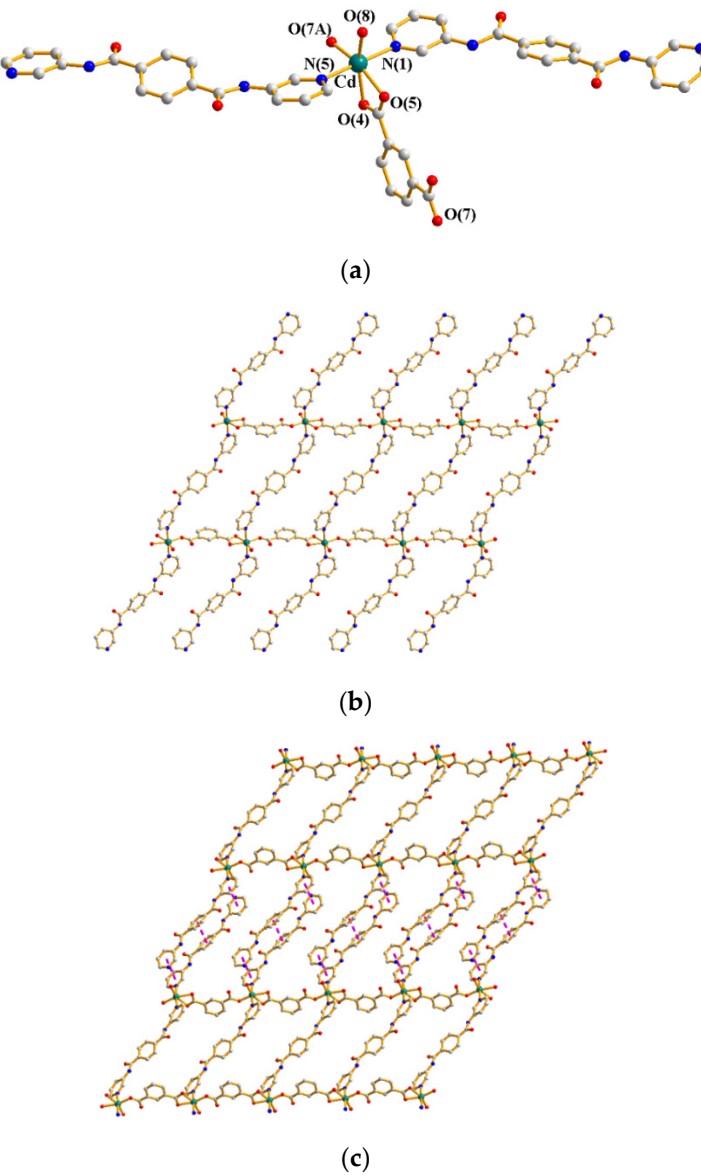

**Figure 3.** (**a**) Coordination environment of Cd(II) ion in **2a**. Symmetry transformations used to generate equivalent atoms: (A) x + 1, y, z. (**b**) A drawing showing the 1D ladder chain with dangling **L** ligands on the sides. (**c**) A drawing showing the 2D layer supported by the π–π interactions (purple dashed lines).

Crystals of **2b** conform to the triclinic space group $P\bar{1}$ and each asymmetric unit consists of two Cd(II) cations, two 1,3-BDC$^{2-}$ ligands, six coordinated water, and four lattice water molecules. Figure 4a depicts a drawing showing the coordination environment of Cd(II), which is seven-coordinated by seven oxygen atoms from two 1,3-BDC$^{2-}$ ligands, three water molecules [Cd-O = 2.277(3) − 2.423(2) Å], resulting in a distorted pentagonal bipyramidal geometry. The Cd(II) cations are further linked together by 1,3-BDC$^{2-}$ to afford 1D chain, Figure 4b. It is noted that solvothermal reaction of Cd(CH$_3$COO)$_2$·2H$_2$O with 1,3-H$_2$BDC in DMF/H$_2$O afforded a 3D framework in which the 1,3-BDC$^{2-}$ ligand bridge four Cd(II) ions [14], while replacement of 1,3-H$_2$BDC with 1,4-H$_2$BDC gave a 1D zigzag chain [14,15]. A comparison with the structure of **2b** indicates that the structures of the Cd(II)-BDC$^{2-}$-based complexes are subject to the donor atom positions of the BDC$^{2-}$ ligands and the solvent system.

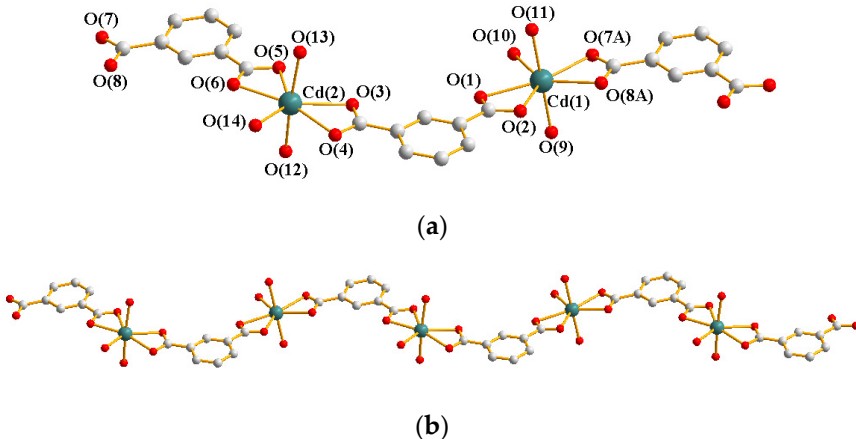

**(a)**

**(b)**

**Figure 4.** (**a**) Coordination environment of Cd(II) ion in **2b**. Symmetry transformations used to generate equivalent atoms: (A) x, y + 1, z − 1 (B) x, y − 1, z + 1 (C) −x + 1, −y + 1, −z. (**b**) A drawing showing the 1D chain.

### 3.3. Structure of 3

The structure of **3** was solved in the triclinic space group $P\bar{1}$ and each asymmetric unit consists of one Cd(II) cation, a half **L** ligand, two halves of a 1,4-BDC$^{2-}$ ligand, two coordinated water, and one lattice water molecules. Figure 5a depicts a drawing showing the coordination environment of Cd(II), which is six-coordinated by five oxygen atoms from two 1,4-BDC$^{2-}$ ligands, two water molecules [Cd-O = 2.2306(18) − 2.3644(17) Å] and one pyridyl nitrogen atom from the **L** ligand [Cd-N = 2.3445(19) Å], resulting in a distorted octahedral geometry. The Cd(II) cations are further linked together by 1,4-BDC$^{2-}$ to afford 1D chains, Figure 5b, which are further linked by the **L** ligands to form a 2D layer. Considering the Cd(II) cations as 3-connected nodes and the organic ligands as linkers, the structure of complex **3** can be simplified as a 3-connected net with the $6^3$-**hcb** topology, Figure 5c, determined by using ToposPro program [12].

### 3.4. Structure of 4

X-ray diffraction analysis reveals that crystals of **4** conform to the monoclinic space group $C2/c$ with each asymmetric unit comprising one Cu(II) cation, one 5-tert-IPA$^{2-}$, and one half **L**. The Cu(II) metal center, Figure 6a, is five-coordinated by four oxygen atoms from four independent 5-tert-IPA$^{2-}$ ligands [Cu-O = 1.962(3) − 2.027(2) Å] and one pyridyl nitrogen atom from the **L** ligand [Cu-N = 2.190(3) Å], resulting in a distorted square pyramidal geometry ($\tau_5$ = 0.006). Dinuclear paddlewheel Cu(II) units are linked by four 5-tert-IPA$^{2-}$ ligands to form 1D looped chains, Figure 6b, with a distance of 2.7002(3) Å between the two Cu(II) ions, which are further linked by the **L** ligands to form a 3D framework. ToposPro program reveals that the structure of **4** can be regarded as a 4-connected net with the ($6^5 \cdot 8$)-**cds** topology, Figure 6c, showing 3-fold interpenetration, Figure 6d, if the dinuclear Cu(II) units are considered as 4-connected nodes.

The structural topology of **4** is the same as that of [Zn(**L**)$_{0.5}$(5-tert-IPA)]$_n$ [9], indicating that the metal identity has no effect on the structural diversity of CPs constructed from the semi-rigid **L** and 5-tert-H$_2$IPA, which is in marked contrast to those from the flexible bpba ligands [6]. For example, reactions of Zn(II) and Cd(II) salts with *N*,*N'*-di(3-pyridyl)adipoamide and 1,2-H$_2$BDC afforded a 1D double-looped chain and a 2D layer, while with 1,3-H$_2$BDC gave a 3-fold interpenetrated **hcb** layers and a 1D ladder chain, respectively [6]. The metal effect on the structural diversity of CPs is thus subject to the flexibility of the bpba spacer ligands.

### 3.5. Ligand Conformation and Bonding Mode

The ligand conformation of **L** can be determined by considering the relative orientation of the two C=O groups, which forms cis and trans if the C=O groups appear in the same

and the opposite direction, respectively [9]. On the other hand, the anti-anti, syn-anti, and syn-syn designations, based on the relative positions of the pyridyl nitrogen and amide oxygen atom, can also be shown [4,16]. The number of possible ligand conformations is thus quite less than that of the flexible bpba ligands with methylene carbon atoms as the spacer [4]. According to the descriptor for **L**, the ligand conformations of **L** in **1**, **2a**, and **3** are trans *anti-anti*, while that in **4** is trans *syn-syn*.

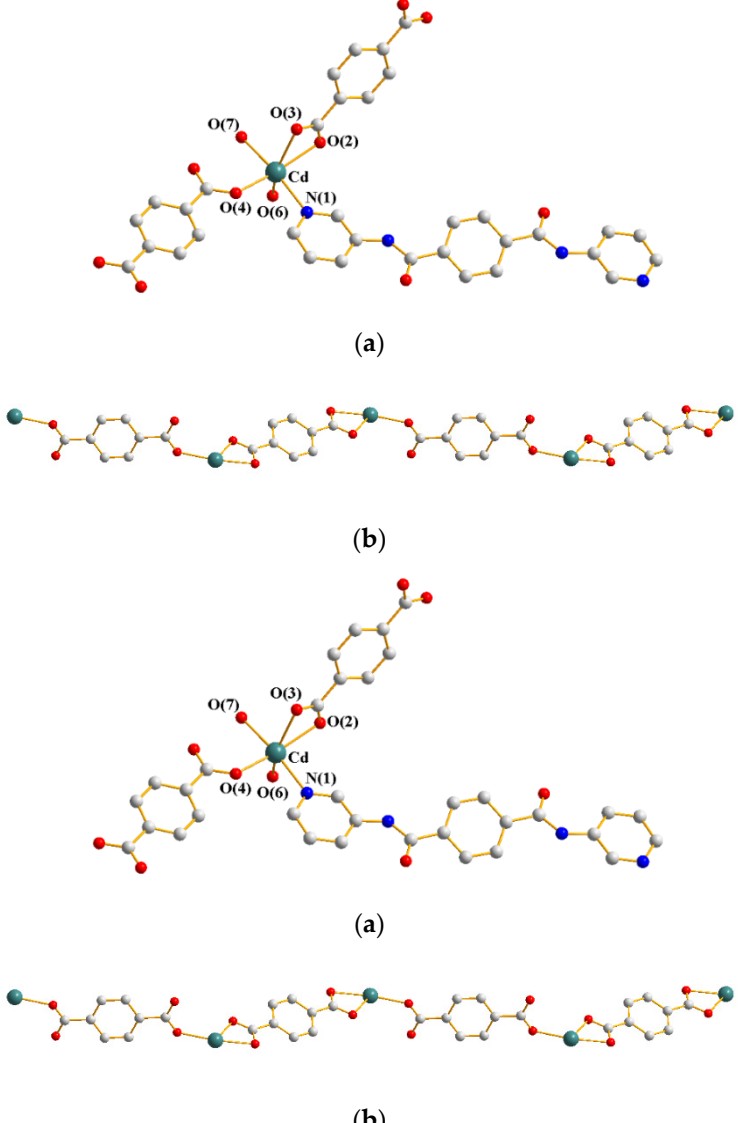

(a)

(b)

(a)

(b)

**Figure 5.** (**a**) Coordination environment of Cd(II) ion in **3**. Symmetry transformations used to generate equivalent atoms: (A) −x, −y + 2, −z + 2 (B) −x + 2, −y + 1, −z + 2 (C) −x + 3, −y + 2, −z + 1. (**b**) A drawing showing the 1D chain supported by the 1,4-BDC$^{2-}$ ligands. (**c**) A drawing showing the 2D layer with the **hcb** topology.

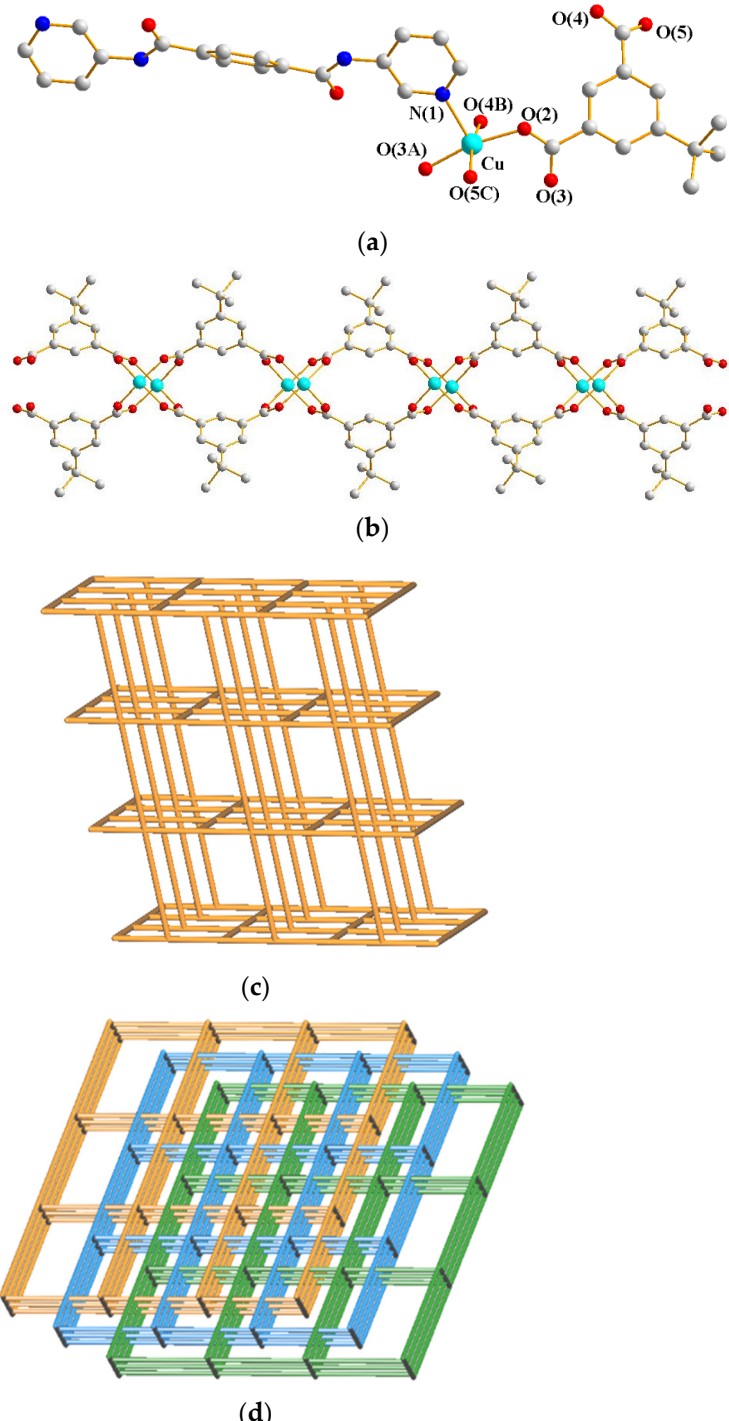

**Figure 6.** (**a**) Coordination environment of Cu(II) ion in **4**. Symmetry transformations used to generate equivalent atoms: (A) −x + 1, −y, −z + 1; (B) −x + 1, y, −z + 1/2; (C) x, −y, z + 1/2. (**b**) A drawing showing the paddlewheel type dinuclear units linked by the 5-tert-IPA$^{2-}$ ligands. (**c**) A drawing showing the 3D net with the **cds** topology. (**d**) A drawing showing the 3-fold interpenetration.

Figure 7 shows the coordination modes for the dicarboxylate ligands. The 1,2-HBTC$^{2-}$ ligands of **1** adopt $\mu_3$-$\kappa^2$O,O′:$\kappa$O″$\kappa^2$O″O‴, coordination mode I, and the 1,3-BDC$^{2-}$ ligands of **2a** and **2b** adopt $\mu_2$-$\kappa^2$O,O′:$\kappa$O″, coordination mode II, and $\mu_2$-$\kappa^2$O,O′:$\kappa^2$O″O‴, coordination mode III, respectively, while the 1,4-BDC$^{2-}$ ligands of **3** display $\mu_2$-$\kappa^2$O,O′:$\kappa^2$O″O‴, coordination IV, and $\mu_2$-$\kappa$O:$\kappa$O′, coordination V, and the 5-tert-IPA$^{2-}$ ligands of **4** adopt $\mu_4$-$\kappa$O,$\kappa$O′:$\kappa$O″$\kappa$O‴, coordination mode VI.

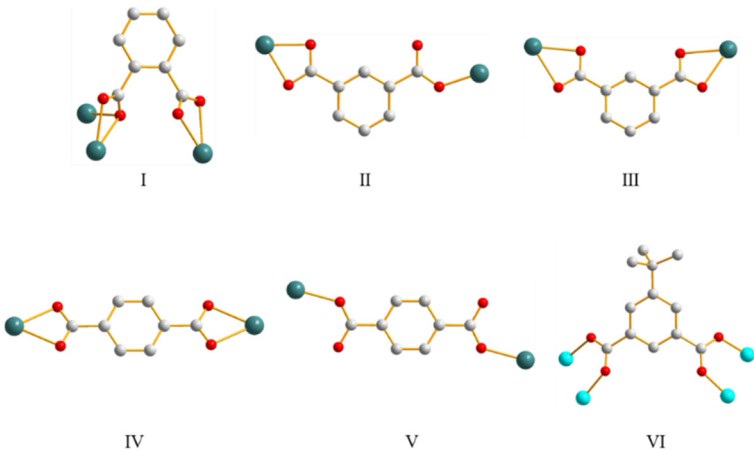

**Figure 7.** Coordination modes for the dicarboxylate ligands.

### 3.6. Ligand Flexibility and Isomeric Effect

Flexible bpba ligands containing different $–(CH_2)_n–$ backbones are susceptible to the changes of the structural diversity because of the ability to adjust to the stereochemical requirements [4]. It has been shown that by manipulating the isomeric polycarboxylic acids, self-catenated CPs based on flexible bpba can be prepared [5–8]. In marked contrast, while the structural diversity of the Cd(II) CPs based on the semi-rigid **L** can be directed by the isomeric $BDC^{2−}$ ligands, only low-dimensional networks of 2D layers and 1D ladder can be found, probably indicating the lack of the flexibility of **L** to adopt the proper conformation for the formation of a self-catenated net. The isomeric effect on the structural diversity of CPs in a mixed system thus relies on the flexibility of the neutral bpba ligands, not to mention the size and geometry of the metal ions.

### 3.7. Thermal Properties

Thermal gravimetric analyses (TGA) were carried out to examine the thermal decomposition of the compounds, Table 2. The samples were recorded from about 25 to 800 °C at 10 °C min$^{-1}$ under a $N_2$ atmosphere. As shown in Figure S4, the TGA curve of **1** shows the gradual weight loss of one coordinated water molecule (calculated 3.97%, observed 4.76%) in 100–280 °C. The weight loss of 81.83% in 280–800 °C corresponds to the decomposition of half **L** and one 1,2-$BDC^{2−}$ ligands (calculated 71.42%). TGA curve of **3**, Figure S5, shows the gradual weight loss of one cocrystallized water molecule (calculated 3.68%, observed 4.12%) in 40–95 °C. The weight loss of two coordinated water molecules (calculated 7.35%, observed 8.21%) in 100–350 °C. The weight loss of 66.41% in 360–706 °C corresponds to the decomposition of one and half **L** and one 1,4-$BDC^{2−}$ ligands (calculated 66.19%). As shown in Figure S6, the TGA curve of **4** shows the weight loss of 81.98% in 250–800 °C corresponds to the decomposition of half **L** and one 5-tert-$IPA^{2−}$ ligands (calculated 85.79%). It is noted that the starting decomposition temperature of the organic ligands of **4** is significantly lower than those of **1** and **3**, probably indicating that the 2D layers of **1** and **3** are more stable than the 3-fold interpenetrated net of **4**.

**Table 2.** Thermal properties of **1**, **3,** and **4**.

| Complex | Weight Loss of Solvent, T, °C (Observed/Calc),% | Weight Loss of Ligand, T, °C (Observed/Calc),% |
|:---:|:---:|:---:|
| **1** | 100–280 (4.76/3.97) | 280–800 (81.83/71.42) |
| **3** | 40–95 (4.12/3.68) 100–350(8.21/7.35) | 360–706 (66.41/66.19) |
| **4** | | 250–800 (81.98/85.79) |

### 3.8. Luminescent Properties

Table 3 summarizes the luminescent properties of **L**, 1,2-H$_2$BDC, 1,4-H$_2$BDC, **1,** and **3**. Figures S7–S11 depict the corresponding excitation/emission spectra, which were measured in the solid state at room temperature. The emissions of **L**, 1,2-H$_2$BDC and 1,4-H$_2$BDC appear in the range of 340–431 nm, which may be ascribed to the intraligand $\pi^* \rightarrow$ n or $\pi^* \rightarrow \pi$ transitions, while **1** and **3** in mixed systems show emissions at 416 and 434 nm upon excitations at 318 and 334 nm, respectively. Since oxidation or reduction of Cd(II) ion is not possible, it is thus not probable that the emissions of **1** and **3** are due to ligand-to-metal charge transfer (LMCT) or metal-to-ligand charge transfer (MLCT) [17]. Therefore, these emissions may be attributed to ligand-to-ligand charge transfer (LLCT). The blue- and red-shift of the emission wavelengths of **1** and **3** with respect to their corresponding organic ligands may be ascribed to the different dicarboxylate ligands that result in different structural types.

**Table 3.** The excitation and emission wavelengths of **L**,1,2-H$_2$BDC, 1,4-H$_2$BDC and complexes **1** and **3** in solid state.

| Ligand. | $\lambda_{ex}$ (nm) | $\lambda_{em}$ (nm) | Complex | $\lambda_{ex}$ (nm) | $\lambda_{em}$ (nm) |
|---|---|---|---|---|---|
| **L** | 276 | 431 | **1** | 318 | 416 |
| 1,2-H$_2$BDC | 283 | 340 | **3** | 334 | 434 |
| 1,4-H$_2$BDC | 277/331 | 384 | | | |

## 4. Conclusions

Divalent CPs constructed from the semi-rigid **L**, polycarboxylic acids, and metal salts have been synthesized successfully under hydrothermal reactions. Complexes **1** and **3** are 2D layers with the **bey** topology and the **hcb** topology, and **2a** and **2b** form a 1D ladder and a zigzag chain, respectively, while **4** is a 3-fold interpenetrated 3D nets with the **cds** topology. The structures of **1–4** are subject to the changes of the polycarboxylate ligands, indicating the effect of the ligand identity as well as the ligand isomerism on the structural diversity. While the flexible bpba are more likely to form self-catenated nets by the manipulation of the isomeric dicarboxylate ligands, the structural diversity is limited for semi-rigid **L**, presumably due to the lack of the flexibility to adopt the proper conformations for the formation of the entangled CPs.

**Supplementary Materials:** The following are available online at https://www.mdpi.com/2624-854 9/3/1/1/s1. Powder X-ray patterns (Figures S1–S3). TGA curves (Figures S4–S6). Emission Spectra (Figures S7–S11). X-ray data for **2a** (Tables S1–S3). Crystallographic data for **1**, **2b**, **3,** and **4** have been deposited with the Cambridge Crystallographic Data Centre, CCDC No. 2046035-2046038.

**Author Contributions:** Investigation, C.-J.C. and C.-L.C.; data curation, Y.-H.L. and J.-H.H.; validation, W.-T.L.; review and supervision, P.M.C. and J.-D.C. All authors have read and agreed to the published version of the manuscript.

**Funding:** This research was funded by Ministry of Science and Technology of the Republic of China, grant number MOST 108-2113-M-033-004.

**Institutional Review Board Statement:** Not applicable.

**Informed Consent Statement:** Not applicable.

**Data Availability Statement:** Data available in a publicly accessible repository.

**Acknowledgments:** We are grateful to the Ministry of Science and Technology of the Republic of China for support.

**Conflicts of Interest:** The authors declare no conflict of interest.

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
