# Peer review of "Coordination Polymers Constructed from Semi-Rigid N,N′-Bis(3-pyridyl)terephthalamide and Dicarboxylic Acids: Effect of Ligand Isomerism, Flexibility, and Identity"

_chemistry, doi:10.3390/chemistry3010001_

Round 1
Reviewer 1 Report
The manuscript reports a detailed crystallographic study of different Cd(II) and Cu(II) complexes obtained by reaction of a metal salt of those ions with a semi-rigid linker, N,N´-bis(3-pyridyl)terephthalamide (L), and different isomeric 1,2-, 1,3- or 1,4-benzenedicarboxylic acid (and 5-tert-butylbenzene-1,3-dicarboxylic acid as well). The influence of the rigidity of the linker and the position of the carboxylate groups of the auxiliary ligands in the polymeric arrangement obtained is described including interesting topologic studies of the nets obtained. The use of the semi-rigid linker (L) limits the structural diversity because it limits the possibility of obtaining self-catenated nets but different nice polymeric arrangements (2D, 1D ladder and zigzag and 3D) were obtained.
In my opinion, this work is interesting mainly from a crystallographic point of view but it may lack of interest for a broader audience. I think that it would be more suitable for publication in another journal such as Crystals or Polymers. In any case, there are some important points that, I consider, need to be addressed before publication.
- PXRD data of 1 shows extra peaks at ca. 2 theta = 25 and 45 deg and PXRD data of 3 shows an extra peak at ca. 9 deg. Do the presence of any starting material explain those peaks?
- According to the experimental section, the crystals were picked after the reaction and not washed with any solvent. Do the presence of a small fraction of the starting materials explain better the elemental analysis of 3 and 4? (Please, check the Calc. Analysis of 4)
- Why the authors do not report the PXRD data of 2? It should show the mixture of 2a and 2b. Did the authors try to modify the reaction conditions to obtain only 2a or 2b?
- I think that section 3.1. is not very useful in its present state. I miss an explanation of the reaction conditions or some aspects related to the IR characterization in section 3.1.
- The authors mention that the single crystal X-ray diffraction data of 2a are humble. The data seems to be good enough to determine the overall 1D ladder chain structure of this compound. However, in my opinion, it must be clearly mentioned in section 3.3 that the quality of the data is not enough to provide a detailed description of the structure. The authors must explain why the structure is not suitable for deposit. According to Table S1, the crystal size and R(int) do not seem to be problematic. Is the structure disordered? Is it twinned? Did the authors try to collect data at low temperature?
-Line 434. TGA curve of 4: The weight loss is clearly in two steps.
Some minor Points:
-Line 18 and 60: Please add: 5-tert-H2IPA = 5-tert-butylbenzene-1,3-dicarboxylic acid
- Line 32 and 33: In my opinion, the 2nd sentence of the paragraph it too specific for the compounds described in this work and not for all the CPs and needs to be rephrased.
- Line 45: Replace 1-3 by 1,3-
- Line 101 and 127: NaOH solutions in water
Reviewer 2 Report
The manuscript reports on the structural diversity achieved by reacting the semi-rigid N,N’-Bis(3-pyridyl)terephthalamide and different dicarboxylic acids with Cd(II) and Cu(II) salts.
The results reported here are new achievements of the authors in the exploration of the reactivity of bis-pyridyl-bis-amide ligands (bpba) in the presence of isomeric dicarboxylate ligands and different metal salts. In particular, the present results are discussed in comparison with previous results obtained using flexible bis-pyridyl-bis-amide ligands and the same dicarboxylic acids.
The work is interesting and the experimental work seems correctly performed. I suggest acceptance after minor revisions according to the following comments.
- To make more evident the influence of various factors (among which the flexibility/rigidity of bpba ligands) on the structural diversity the authors could add to the text a new table which summarize the previous results in comparison with the new ones.
- Line 18: define the acronym 5-tert-IPA in the formula of compound 4.
- Lines 22-24: Rephrase to make more clear the aim of the work.
- Line 46: change “3D net cds topology “ with “3D net of cds topology.”
- Lines 149-155: use bold characters to indicate the number of compounds and L.
- Powder diffraction pattern for compound 1 (Fig S1) seems to have some extra peaks at about 22° and 47°. Please, comment.
- Line 160: “…..coordination environment about 1, which….” should be “…..coordination environment about Cd(II), which….”. This comment applies to all the other structural descriptions.
- Line 249: The description of the AU for 2b do not match with figure 4a and with the crystal data. Actually, it contains two Cd(II) atoms and two 1,3-BDC2- ligands and water molecules. Please, check carefully and correct.
- Line 280: “….the coordination environment about 2,…..”, please correct.
- Line 426: change the word complexes with compounds
Reviewer 3 Report
The ms of Chen, Chen and other describes some new cadmium coordination polymers based on semi-rigid N,N’-bis(3-pyridyl)terephthalamide and polycarboxylate ligands (benzene-1,3-dicarboxylic, benzene-1,4-dicarboxylic acid and, benzene-1,2-dicarboxylic acids). It is very difficult to consider the ms in this form suitable for Chemistry. A basic experimental data are missing in the manuscript. The elemental analysis of a compound 3 introduces misleading data and are inconsistent with the proposed formula. Moreover, more detailed spectroscopic characterization, the elemental analysis should be implemented, PXRD patters of as-synthesized compounds 2a and 2b should be included and described. The sentence “Different colorless crystals with poor crystallinity were obtained, which were difficult to be separated manually” is strange to me. Did the authors try to separate the obtained compounds in a way other than manual?
Round 2
Reviewer 1 Report
The authors have kindly addressed all my concerns. Therefore, I agree with the publication of the article.
Reviewer 3 Report
In my opinion, this work should be publish after completing section 2.3.2 (e.g. introduction of the IR data of compounds 2a and 2b).
